

# The FengYun-3C radio occultation sounder GNOS: a review of the mission and its early results and science applications

**Yueqiang Sun[1,2,3], Weihua Bai[1,2,3], Congliang Liu[1,2], Yan Liu[4], Qifei Du[1,2,3], Xianyi Wang[1,2], Guanglin Yang[5], Mi Liao[5], Zhongdong Yang[5], Xiaoxin Zhang[5], Xiangguang Meng[1,2], Danyang Zhao[1,2], Junming Xia[1,2], Yuerong Cai[1,2], and Gottfried Kirchengast[6,2,1]**

[1] National Space Science Center, Chinese Academy of Sciences (NSSC, CAS) and Beijing Key Laboratory of Space Environment Exploration, Beijing, China

[2] Joint Laboratory on Occultations for Atmosphere and Climate (JLOAC) of NSSC, CAS, Beijing, China, and University of Graz, Graz, Austria

[3] School of Astronomy and Space Science, University of Chinese Academy of Sciences, Beijing, China

[4] National Meteorological Center, Chinese Meteorological Administration, Beijing, China

[5] National Satellite Meteorological Center, Chinese Meteorological Administration, Beijing, China

[6] Wegener Center for Climate and Global Change (WEGC) and Institute for Geophysics, Astrophysics, and Meteorology/Institute of Physics, University of Graz, Graz, Austria

Correspondence to: Congliang Liu (Email:liucongliang1985@gmail.com); Wehuai Bai (Email: bjbwh@163.com)



**Abstract**

The Global Navigation Satellite System (GNSS) occultation sounder (GNOS) is one of the new generation payloads onboard the Chinese FengYun 3 (FY-3) series of operational meteorological satellites for sounding the Earth's neutral atmosphere and ionosphere. FY-3C GNOS, onboard the FY-3 satellite C launched in September 2013, was designed for acquiring setting and rising radio occultation (RO) data by using GNSS signals from both the Chinese BeiDou System (BDS) and the U.S. Global Positioning System (GPS). So far, the GNOS measurements and atmospheric and ionospheric data products have been validated and evaluated and then been used for atmosphere and ionosphere related scientific applications.

This paper reviews the FY-3C GNOS instrument, RO data processing, data quality evaluation, and research applications. The reviewed data validation and application results demonstrate that the FY-3C GNOS mission can provide accurate and precise atmospheric and ionospheric GNSS (i.e., GPS and BDS) RO profiles for numerical weather prediction (NWP), global climate monitoring (GCM) and space weather research (SWR). The performance of the FY-3C GNOS product quality evaluation and scientific applications establishes confidence that the GNOS data from the series of FY-3 satellites will provide important contributions to SWP, GCM and SWR scientific communities.

# 1 Introduction

The Global Navigation Satellite System (GNSS) radio occultation (RO) technique (Melbourne et al., 1994; Ware et al., 1996) for sounding the Earth's neutral atmosphere and ionosphere was demonstrated by the proof-of-concept Global Positioning System/Meteorology (GPS/MET) mission launched in 1995 (Ware et al., 1996; Kursinski et al., 1996; Kuo et al., 1998), and the



following GNSS RO missions such as the Challenging Mini-satellite Payload (CHAMP) (Wickert et al., 2001, 2002), the Constellation Observing System for Meteorology, Ionosphere and Climate (COSMIC) (Anthes et al., 2000, 2008; Schreiner et al., 2007), the Gravity Recovery and Climate Experiment (GRACE) (Beyerle et al., 2005; Wickert et al., 2005), and

the Meteorological Operational (MetOp) satellites (Edwards and Pawlak, 2000; Luntama et al., 2008). These missions have demonstrated the unique properties of the GNSS RO technique, such as high vertical resolution, high accuracy, all-weather capability and global coverage (Ware et al., 1996; Gorbunov et al., 1996; Rocken et al., 1997; Leroy, 1997; Steiner et al., 1999), and long-term stability and consistency of different RO mission observations (Foelsche et al.,

2009, 2011). Therefore, GNSS RO data products (i.e., bending angle, refractivity, temperature, pressure, water vapor, and ionospheric electron density profiles) have been widely used for numerical weather prediction (NWP) (e.g., Healy and Eyre, 2000; Kuo et al., 2000; Healy and Thepaut, 2006; Aparicio and Deblonde, 2008; Cucurull and Derber, 2008; Poli et al., 2008; Huang et al., 2010; Le Marshall et al., 2010; Harnisch et al., 2013), global climate monitoring

(GCM) (e.g., Steiner et al., 2001, 2009, 2011, 2013; Schmidt et al., 2005, 2008, 2010; Loescher and Kirchengast, 2008; Ho et al., 2009, 2012; Foelsche et al., 2011a; Lackner et al., 2011) and space weather research (SWR) (Anthes, 2011; Anthes et al., 2008; Arras et al., 2008; Brahmanandam et al., 2012; Pi et al., 1997; Wickert, 2004; Yue et al., 2015).

The development of GNSS such as China's BeiDou navigation satellite system (BDS), Russia's

Global navigation satellite system (GLONASS), and the European Galileo system, has significantly enhanced the availability and capacity of the GPS-like satellites, which will make GNSS RO even more attractive in the future (Bai et al., 2017). These new GNSS satellites, together with planned low Earth orbit (LEO) missions, will offer much more RO observations



in future (Wang et al., 2015; Cai et al., 2017). One of these LEO missions is China's GNSS occultation sounder (GNOS) onboard the FengYun 3 series C (FY-3C) satellite, which was successfully launched on 23 September 2013 (Wang et al., 2013, 2014; Bai et al., 2014b, 2017; Liao et al., 2015, 2016a, 2016b; Wang et al., 2015; Du et al., 2016).

The FY-3C GNOS mission was designed and developed by National Space Science Center, Chinese Academy of Sciences (NSSC, CAS), for sounding the Earth's neutral atmosphere and ionosphere by using both the BDS and GPS signals (Wang et al., 2015; Bai et al., 2017). The FY-3C satellite is the first BDS/GPS compatible RO sounder with a state-of-the-art RO receiver ─GNOS (Bai et al., 2012, 2014a; Wang et al., 2015; Du et al., 2016). The following FY-3

series of operational meteorological satellites (Figure 1) will continue to carry GNOS as a major payload. The next one of these satellites is FY-3D, which will be launched later in 2017 (Liao et al, 2016; Yang, et al, 2017; Sun et al., 2017).

The FY-3C GNOS instrument consists of three antennas, three radio frequency units (RFUs) and an electronic unit (EU), which uses high-dynamic, high-sensitivity signal acquisition and

tracking techniques (Figure 2). The three RFUs are installed close to their corresponding antennas by using sharp cavity filters, to reduce the loss of the cable between antennas and RFUs. The EU is the major component of GNOS, which accomplishes the GNSS remote sensing signals acquisition and tracking as well as the real-time positioning and carrier phase observations (Wang et al., 2015). In FY-3C GNOS design, the different features of BDS and

GPS signals have been taken into account, and it can observe both the neutral atmosphere and ionosphere by using both BDS and GPS signals.

As shown in Figure 2, the FY-3C GNOS instrument involves a positioning antenna, a rising occultation antenna, and a setting occultation antenna as part of its physical structure. In term of

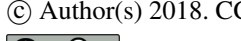



electrical structure, the FY-3C GNOS involves five antennas because each occultation antenna has an ionosphere occultation antenna and an atmosphere occultation antenna (Bai et al., 2014).

The positioning antenna is a wide beam, low-gain antenna pointing toward zenith, which can track six BDS and eight GPS satellites, simultaneously. Its measurements are used for real-time navigation, positioning and the LEO satellites' precise orbit determination (POD) in post processing.

The front-view (along the satellite velocity direction) occultation antenna and back-view (satellite anti-velocity direction) occultation antenna are used for rising and setting occultation event tracking, respectively. The FY-3C GNOS can track four BDS and six GPS occultation events simultaneously. The atmosphere occultation antennas have a pattern that is wide in azimuth and narrow in elevation. A gain of approximate 10 dBi is reached over the coverage range between about ±35 degree in azimuth and between about ±7.5 degree in elevation (Bai et al., 2014; Du et al, 2016).

The EU of FY-3C GNOS is based on a field-programmable gate array (FPGA) and digital signal processor (DSP) framework. After filtering and down-conversion in the RFU, the signals are digitally down converted with an analog-to-digital converter (ADC), then sampled at a high rate and transmitted to the channel processor of the EU, where the GNOS accomplishes navigation, positioning and occulting GNSS satellite prediction and selection, signal acquisition and tracking, and data handling. An ultra-stable oscillator (USO) is used as a reference oscillator with highly stable frequency (1-s Allan deviation of $10^{-12}$), in order to retrieve atmospheric measurements with high accuracy (Du et al., 2016; Sun et al., 2017). It also allows using the zero-difference method to invert the excess phase measurements (Beyerle et al., 2005; Bai et al., 2017).



The FY-3C GNOS is a multi-frequency receiver with BDS/GPS compatibility, BDS B1/B2 closed-loop tracking, GPS L2 codeless-mode operation for P code, GPS L2C closed-loop tracking and GPS L1 C/A closed-loop and open-loop tracking capabilities. The capability for both BDS and GPS increases the number of transmitting sources and promises significant

enhancements in total throughput of measurements. The FY-3C GNOS receiver measures the following observable parameters: for each tracked GPS satellite, L1 C/A-code phase, L1 carrier phase, L1 signal amplitude, L2 P-code phase, L2C code phase (if present), L2 carrier phase, L2 signal amplitude; and for each tracked BDS satellite, B1I code phase, B1 carrier phase, B1 signal amplitude, B2I code phase, B2 carrier phase, and B2 signal amplitude (Bai et al., 2014).

In the lower part of the troposphere, where highly dynamic signal conditions are frequently encountered due to the strong atmospheric modulations, the GPS L1 signal is tracked in open loop in parallel with the closed-loop tracking. In open-loop tracking, the signal is down-converted using a numerically controlled oscillator, which generates a frequency given by an onboard Doppler model pre-calculated in GNOS without a feedback from received signal

(Sokolovskiy, 2001; Sokolovskiy et al., 2009). Particularly, for the rising occultation, an a priori range model of the atmospheric delay (Ao et al., 2009) is also calculated on board the GNOS. The baseband signal is then sampled at a rate of 100 kHz. Furthermore, a sampling rate of 100 Hz of open-loop tracking is proven to be sufficient to capture the signal modulated by the atmosphere dynamics and uncertainties of the Doppler model. The designed parameters of

FY-3C GNOS are summarized in Table 1; it can be seen that some parameters of the FY-3C GNOS are comparable to those of COSMIC (Rocken et al., 2000) or MetOp/GRAS (Loiselet et al., 2000).



After this introduction to the topic and the GNOS instrument, the paper is structured as follows. Section 2 provides a description of FY-3C GNOS data processing and data products. Section 3 describes the validation and evaluation of FY-3C GNOS data. Section 4 presents the GNOS RO data applications. Finally, a summary and conclusions are given in Section 5.

## 2 FY-3C GNOS data processing and data products

The FY-3C satellite flies in a sun-synchronous polar orbit of inclination $98.8^{\circ}$, mean altitude 836 km and orbital period 101.5 minutes. Therefore, GNOS can observe the troposphere, stratosphere and ionosphere from the Earth surface to around 800 km altitude. As a multi-GNSS receiver, GNOS has the capability of observing the phases and amplitudes of radio waves from GPS and BDS satellites as they are occulted by the Earth's atmosphere. From the raw GNOS observations, accurate and precise vertical bending angle profiles are obtained in the troposphere, stratosphere and ionosphere. Based on the bending angles, profiles of atmospheric/ionospheric refractivity are calculated. The refractivity is a function of temperature, pressure, water vapor pressure, and electron density, so the refractivity profiles can be used to derive profiles of temperature and water vapor in the troposphere, temperature in the stratosphere and electron density in the ionosphere. The operational processing procedure and the product levels of the FY-3C GNOS RO data products are shown in Figure 3.

### 2.1 FY-3C GNOS mission system architecture

As shown in Figure 3, the FY-3C GNOS mission system consists of three major segments, i.e., GNSS satellites segment, LEO satellite segment, and ground segment. The GNSS segment is composed of the GPS system and the (currently still regional) BDS system. The latter includes 5

geostationary orbit (GEO) satellites, 5 inclined geosynchronous orbit (IGSO) satellites and 4

medium earth orbit (MEO) orbit satellites, available to conduct radio occultation. The LEO

satellite segment is composed of the FY-3C satellite carrying the GNOS RO receiver. The

ground segment involves a data processing center in Beijing and 5 ground stations, located in

Beijing, Wulumuqi, Guangzhou, Jiamusi and Kiruna, respectively. In addition, in the ground

segment, auxiliary information provided by the international GNSS service (IGS) stations, such

as the GPS/BDS precise orbits, clock files, Earth orientation parameters, and the coordinates

and measurements of the ground stations, are also needed.

## 2.2 FY-3C GNOS Level-0 data preparation

In total there are 12 instrument payloads onboard the FY-3C satellite and their observed data is

downloaded to the ground data stations, and then transferred to the data processing center in

Beijing. In the data processing center, firstly the raw observed data are decrypted and

decompressed, and then the data packets are classified and stored in 12 different specific storage

spaces, according to the number of different instrument payloads. One of the 12 data packets is

the binary format FY-3C GNOS observations, mainly including phase and SNR (signal to noise

ratio) measurements, and these raw data are defined as the Level-0 data.

## 2.3 FY-3C GNOS Level-1 data processing

The unpacking and interpretation program module verifies and revises the data package format

of the FY-3C Level-0 raw data; and then unpacks them into 7 packages, i.e., GPS positioning

package, GPS ionospheric occultation package, GPS atmospheric closed-loop occultation

package, GPS atmospheric open-loop occultation package, BDS positioning package, BDS





ionospheric occultation package, and BDS atmosphere occultation package; finally the progam

processes and stores the data according to the naming rules and data storage principles. The

format conversion program module subsequently converts the data into receiver-independent

exchange format (RINEX) or NetCDF format. According to the time period of the observed

data, the auxiliary data acquisition program module automatically downloads the BDS/GPS

ephemeris and almanac data, from the IGS website and other specific websites, for use by the

LEO precise orbit determination (POD) module.

Highly accurate measurements by the GNSS and LEO satellites in terms of time and position

are the key to successful retrieval of the atmospheric and ionospheric profiles of an occultation

event. According to the GNSS positioning package data and satellite precision ephemeris data,

the POD program module conducts the LEO POD to obtain LEO satellite accurate position,

velocity, satellite attitude, receiver clock error and clock drift information. Based on the

measurements of pseudo range and carrier phase as well as the attitude information of the

GNOS POD antenna, the GNSS clock offsets, GNSS precise orbit information, and the Earth

orientation parameters, the LEO POD can be conducted by integrating the equations of celestial

motion. Currently, in FY-3C GNOS data processing, the Bernese software (V5.0) (Dach et al.,

2007) and the position and navigation data analyst (PANDA) software (Zhao et al, 2017) have

been used and evaluated, and both of them can deliver highly accurate POD results.

Based on the LEO POD data and occultation observations, the excess phase calculation program

module is run to determine excess phase data by using the single- or zero-differencing method

(Bai et al., 2017). Generally, in the processing of GNOS data, the single-differencing technique

is applied to obtain the excess phase as a function of time in an Earth-centred inertial reference

frame. Under the condition of fewer reference satellites, a zero-difference technique should be



more appropriate for BDS, since it does not require a reference satellite for simultaneous observations but requires an ultra-stable oscillator on an LEO receiver (Beyerle et al., 2005), which is available for GNOS (cf. Table 1; Bai et al., 2017).

## 2.4 FY-3C GNOS Level-2 data processing

The data pre-processor module includes the atmospheric and ionospheric occultation data pre-processing modules, which mainly involves quality control, data filtering, and open loop data quality control and initial processing.

The atmosphere inversion module includes the atmospheric impact parameter and bending angle calculation, inversion to refractivity, and temperature, pressure and humidity retrieval.

The radio occultation processing package (ROPP) software (V6.0) developed at ROM SAF (radio occultation meteorology satellite application facility) is used for this purpose. More specifically, from the excess phase the Doppler frequency can be obtained, then the bending angles are determined from the Doppler frequency shift and the corresponding satellite positions and velocities (e.g., Kursinski et al., 1997). In order to retrieve neutral atmospheric parameters,

the ionosphere effects on the bending angles need to be eliminated. For the GNSS L band signals, the ionosphere refractivity is proportional to the inverse square of the frequencies, whereas the neutral atmosphere refractivity is almost independent of the frequencies (Vorob'ev and Krasil'nikova, 1994; Syndergaard, et al., 2000). Therefore, a dual-frequency linear combination can mostly correct the first-order ionospheric effects (Vorob'ev and Krasil'nikova,

1994). However, there are some higher-order ionospheric effects that still remain in the bending angle profiles (Kursinski et al., 1997; Liu et al., 2013, 2015, 2016, 2017a). To reduce the ionospheric residual errors and other small-scale noise, the statistical optimization technique is





used together with the MSISE-90 climatology model. An optimal linear combination is expressed as a matrix equation to compute the neutral atmospheric bending angle and the ionospheric bending angle.

After ionospheric correction, under the assumption of local spherical symmetry, the refractive index can be retrieved by an Abel transform from a given bending angle profile $\alpha$ as function of impact parameter $a$, as shown in Eq. (1) (Fjeldbo et al., 1971; Melbourne et al., 1994; Kursinski et al., 1997), and then, refractivity $N$ can be obtained from the refractive index $n$ as shown in Eq. (2).

$$n(a_0) = \exp\left[\frac{1}{\pi}\int_{a_0}^{\infty}\frac{\alpha(a)}{\sqrt{a^2 - a_0^2}}da\right] \tag{1}$$

$$N = (n-1)\times 10^6 \tag{2}$$

The refractivity is a function of temperature ($T$), pressure ($p$), water vapor pressure ($e$), and electron density ($n_e$), as shown in Eq. (3).

$$N = 77.6\frac{p}{T} + 3.73\times 10^5\frac{e}{T^2} - 4.03\times 10^7\frac{n_e}{f^2} \tag{3}$$

where $f$ is the frequency of the GNSS signal. Therefore the refractivity profiles can be used to derive profiles of temperature, water vapor and ionospheric electron density.

Because of the ambiguity of temperature and humidity in lower troposphere (Healy and Eyre, 2000; Poli et al., 2002), one-dimensional variation (1-D-Var) analysis, involving co-located profiles of the Chinese global forecast model (used as background at T639L60 resolution), is used to retrieve temperature and humidity profiles (Liao et al., 2016).





The ionosphere inversion module involves the ionospheric Total Electron Content (TEC) calculation and the electron density retrieval, in which the dual frequency difference method is used. The FY-3C GNOS ionospheric occultation data mainly include dual frequency carrier phase and SNR observations, with a sampling rate of 1 Hz. Since the primary mission of FY-3C

5    GNOS is the neutral atmosphere occultation sounding, when there is no free channel, a new atmospheric occultation event will occupy an ionospheric occultation channel. Therefore, the number of complete ionospheric occultation profiles is less than that of the atmospheric RO events, and there are typically around 220 GPS and 130 BDS ionospheric RO events per day.

When GNSS signals transmitted through the ionosphere from GNSS satellites to the FY-3C

10    satellite are bent and delayed by ionosphere refraction effect, the TEC can be calculated by using Eq. (4) (e.g., Syndergaard et al., 2000),

$$TEC = \frac{f_1^2 f_2^2}{C(f_1^2 - f_2^2)}(L_1 - L_2),$$  (4)

where $L_1$ and $L_2$ are the dual-frequency carrier phase observations, constant C=40.3082 m$^3$ s$^{-2}$, and $f_1$ and $f_2$ are the two signal frequencies. Then the electron density $N_e$ in the

15    ionosphere is derived by using an inverse Abel transformation, as shown in Eq. (5),

$$N_e(r) = -\frac{1}{\pi}\exp(\int_{r_0}^{LEO} \frac{dTEC/dr_0}{\sqrt{r_0^2 - r^2}}dr_0),$$  (5)

where $N_e$ is electron density and $r$ the impact parameter for ionospheric altitudes.



## 3 Preliminary validation and evaluation of the FY-3C GNOS data

### 3.1 Evaluation of the FY-3C satellite POD data

The FY-3C GNOS observations and its POD results, e.g., clock estimates, position and velocity, are fundamental and crucial elements of the whole GNOS RO data processing chain. Thus, the

FY-3C satellite POD performance has been evaluated by difference GNSS data processing centers (Liao et al., 2016; Li et al., 2017; Xiong et al., 2017; Zhao et al., 2017) by using the Bernese software (Liao et al., 2016) and PANDA software (Zhao et al., 2017), separately. In all these POD data quality evaluations, an internal consistency metrics method named overlapping orbit differences (OODs) (Zhao et al., 2017) has been used.

The Li et al. (2017) study showed the overlapping orbit consistency, and the 3D RMS of OODs was found 2.7 cm by using GPS data only and 3.4 cm by using both GPS and BDS data together. Furthermore, the 3D RMS of OODs was found 30.1 cm by using all the BDS data, and 8.4 cm by using the MEO and IGSO BDS data only. Similarly, the Xiong et al. (2017) study also showed overlapping orbit consistency, and the 3D RMS of OODs was found about 3.8 cm by

using both GPS and BDS data together, and about 22 cm by using BDS data only.

Zhao et al. (2017) used the GPS and BDS observed data to conduct the POD of the FY-3C satellite, respectively. The solutions showed that the 3D RMS of 6-h OODs reached 2.3 cm and 15.8 cm for GPS-only and BDS-only procedures, respectively. The quality of the FY-3C POD result calculated by using the BDS data is worse than that by using GPS data mainly because of

the limited number of BDS satellites, and lower quality of BDS satellite POD data due to small number of ground stations as well as restricted distribution of ground stations. In the same study,





Zhao et al. (2017) improved the quality of BDS satellite POD data by using ground stations data and accurate FY-3C POD data, which were retrieved from GPS observations.

In NSSC, both the Bernese and PANDA software have been used in the FY-3C satellite POD processing and in this paragraph the POD performance of PANDA software is briefly discussed.

Since the BDS is an incomplete constellation and can only provide regional navigation and positioning services, mainly in Asia-Pacific area, the FY-3C GNOS satellite POD processing is implemented by using the reduced-dynamic orbit determination method in which the zero-differencing algorithm with a least-squares estimator method and dual-frequency ionospheric correction has been conducted (Cai et al., 2017). The quality analysis of the FY-3C

POD has been conducted by using two months of data. The results showed that the mean RMS of 6-hour OODs along radial, tangential, and normal directions are 1.24 cm, 1.60 cm, and 3.07 cm, respectively (Cai et al., 2017).

## 3.2 Validation and evaluation of the FY3 C-GNOS atmospheric profiles

After the launching of the FY-3C satellite, an in-orbit testing of the FY-3C GNOS data has been

presented by several papers (Wang et al., 2015; Liao et al., 2017; Bai et al., 2017).

In the Wang et al. (2015) study the GNOS RO events and their global distribution were analyzed; comparing with the GPS RO observations, the accuracy and consistency of BDS real-time positioning results and BDS RO products were analyzed. The preliminary results showed that comparing with the number of GPS GNOS RO events, the regional incomplete

BDS system with 14 navigation satellite can enhance the number of RO events by about 33% (Figure 4). The statistical BDS and GPS GNOS RO data analyses, by using 17 pairs of BDS/GPS GNOS RO events in a week, showed that the BDS/GPS difference standard deviation



of refractivity, temperature, humidity, pressure and ionospheric electron density are lower than 2 %, 2 K, 1.5 g/kg, 2 %, and 15.6 %, respectively. Therefore, the BDS observations/products are in general consistent with those from GPS (Wang et al., 2015).

Comparing with the co-located ECMWF (European Centre for Medium-Range Weather Forecasts) analysis model data, the quality of atmospheric refractivity profiles from the FY-3C GNOS mission have been evaluated by Liao et al. (2016). The results (Figures 5 and 6) showed that the mean bias of the refractivity obtained through GPS (BDS) GNOS was approximately –0.09 % (–0.04 %) from the near surface to about 45 km. While the average standard deviation was approximately 1.81 % (1.26 %), it was as low as 0.75 % (0.53 %) in the range of 5–25 km, where best sounding results are usually achieved.

Since FY-3C GNOS uses an ultra-stable oscillator with 1-sec stability (Allan deviation) at the level of $10^{-12}$, both zero-differencing and single-differencing excess phase processing methods are basically feasible for FY-3C GNOS observations. Focusing on the evaluation of the bending angle and refractivity profiles that derived from zero-differencing and single-differencing excess phase data of BDS RO, a comparison analysis has been conducted by using a 3-month set of GNOS BDS RO data (October to December 2013) and the co-located profiles from ECMWF analyses (Bai et al., 2017). The statistics showed that the results from single- and zero-differencing are consistent in both bias and standard deviation, also demonstrated the feasibility of zero-differencing for GNOS BDS RO observations. The average bias (and standard deviation) of the bending angle and refractivity profiles were found to be as small as about 0.05 % to 0.2 % (and 0.7 % to 1.6 %) over the upper troposphere and lower stratosphere, including for the GEO, IGSO, and MEO subsets (Bai et al., 2017).





### 3.3 Validation and evaluation of the FY3 C-GNOS ionospheric profiles

We evaluated the FY-3C GNOS ionosphere electron density profiles through a statistical
comparison analysis with ionosonde data, in which both the BDS and GPS GNOS RO data
were compared against observations from 69 ionosonde stations. For detailed information of
FY-3C GNOS ionosphere occultation data processing, and the evaluation analysis algorithms,
we refer to Yang et al. (2017). Relevant statistical results are shown in Figure 7. The linear
regression of absolute NmF2 values derived from the GNOS GPS occultation and ionosonde
data, give a correlation coefficient of 0.95, statistical bias of 3.0 %, and standard deviation of
17.9 %, in which a total of 547 matching pairs of data were used. Similarly, the linear
regression of absolute NmF2 values derived from the GNOS BDS occultation and ionosonde
data, gives a correlation coefficient of the fitted regression is 0.95, statistical bias of 4.7 %, and
standard deviation of 19.2 %, in which a total of 376 matching pairs of data were used. One can
see that the bias and standard deviation of the NmF2 derived from the GNOS BDS occultation
and GPS occultation are consistent and comparable (Yang et al., 2017).

### 4 Applications of FY-3C GNOS data products

### 4.1 Applications of the FY3 C-GNOS atmospheric products

The FY-3C GNOS data, mainly including bending angle and refractivity profiles, have been
assimilated in the global/regional assimilation and prediction system (GRAPES) of the China
Meteorological Administration (Liu et al., 2014), for the NWP application. Figure 8 shows the
effects of the GPS GNOS RO data only (red line) and both the BDS and GPS GNOS RO data
(blue line), used in 7-day forecast at 500 hPa altitude level, on the NWP results, in which the

black line denotes the reference line, as well as the x-axis and y-axis denote the forecast time

range and the anomaly correlation coefficient (ACC), respectively. The results indicated that

FY-3C GNOS RO data have a positive effect on analysis and forecast at all medium-term

forecast ranges in GRAPES, not only in the southern hemisphere where conventional

observations are lacking but also in the northern hemisphere where data are more dense.

Figure 9 shows an evaluation score card of the effects of the GPS and BDS FY-3C GNOS RO

data on the GRAPES forecast results, in which the grey color denotes the effect is tiny, the red

color denotes positive effect and the green color denotes negative effect. The evaluation results

showed that except somewhere in north hemisphere and East Asia, the GPS and BDS GNOS

data trend to have overall clearly positive effects on the GRAPES NWP results. Since June

2017, the GNOS RO data are therefore operational assimilated into GRAPES by the CMA.

## 4.2 Applications of the FY3 C-GNOS ionospheric products

The FY-3C GNOS ionospheric products have been applied in the ionospheric and space

weather research areas. A comparison has been made between ionospheric peak parameters

retrieved by FY-3C GNOS RO data and those measured by globally distributed ionosondes

(Mao et al., 2016). Reasonable agreement was obtained in this case; the results indicated that

NmF2 and hmF2 retrieved from FY-3C GNOS measurements are reliable and can be used for

ionospheric physics studies. The comparison between the FY-3C GNOS data and the IRI model

is also reasonably good, but the IRI model tends to overestimate NmF2 at the crests of the

equatorial anomalies (Mao et al., 2016). In addition, Yang et al. (2016) analyzed sporadic

E-layer events, caused by precipitating particle in the auroral region, by using FY-3C GNOS

observations. The results showed that the disturbance intensity of the sporadic E-layers in the

summer hemisphere is significantly greater than that in the same latitude region in the winter hemisphere. Also the occurrence rate of sporadic E-layers in summer hemisphere was found significantly higher than that the winter hemisphere.

Based on the validated FY-3C GNOS ionospheric data and ionosonde stations observations, the
global ionospheric effects of the strong magnetic storm event in March 2015 were analyzed by Bai et al. (2017b). Both the analyses of GNOS ionospheric data and ionosonde station observations showed that the magnetic storm caused a significant disturbance in NmF2 and hmF2 levels, and the average NmF2 featured the same basic trends in the zone of geomagnetic inclination between 40 ° and 80 °. Suppressed daytime and nighttime NmF2 levels indicated
mainly negative storm conditions (Figure 10). The analysis in this way also demonstrated the value of the FY-3C GNOS data and especially confirmed the utility of its ionosphere products for statistical and event-specific ionospheric physics analyses.

## 5 Summary and Conclusions

FY-3C GNOS is the first BDS/GPS-capable GNSS RO sounder in space and combines a
state-of-the-art RO receiver, which has meanwhile been successfully running in orbit for more than four years. So far, a large dataset of FY-3C GNOS RO observations has been obtained, which includes both atmospheric and ionospheric RO products. These products have been validated and evaluated by difference institutions and the results showed that FY-3C satellite POD accuracy reaches centimeter level.

Regarding the GNOS atmospheric data, comparing with co-located ECMWF analysis data, the
mean bias (standard deviation) of the atmospheric refractivity is less than 1 % (2 %), in the upper troposphere and lower stratosphere. Comparing the GNOS ionospheric data with

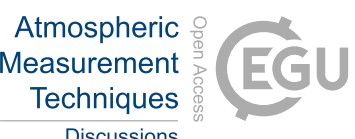

ionosonde station data, the statistical bias (standard deviation) of both the BDS and GPS GNOS

ionospheric electron density NmF2 values is less than 5 % (20 %). The consistency of BDS and

GPS RO products has been successfully validated as well; it has proven the utility of the GNOS

RO data for applications in numerical weather prediction (NWP), global climate monitoring

(GCM) and space weather research (SWR).

So far the GNOS RO data products have been widely used in NWP and SWR applications in

China mainly. In the near future, the GNOS RO data will also be heavily used for GCM

applications, through cooperation research between NSSC (Beijing, China) and WEGC (Graz,

Austria). Wider international use in NWP and other applications will also follow based on the

continuously improvement data product quality of the GNOS data.

With the further expansion of the GNSS transmitter satellite constellations, and the additional

GNOS instruments launched onboard the FY-3 series of satellites, the FY-3 GNOS observations

are expected to provide an essential future contribution to the pool of international high-quality

atmospheric and ionospheric RO products, for substantial benefit of NWP, GCM and SWR.

**Acknowledgements**

This research was supported by the National Natural Science Foundation of China (grant Nos.

41405039, 41775034, 41405040, 41505030 and 41606206), the Strategic Priority Research

Program of the Chinese Academy of Sciences (grant No. XDA15012300), the Scientific

Research Project of the Chinese Academy of Sciences (grant No. YZ201129), and the FengYun

3 (FY-3) Global Navigation Satellite System Occultation Sounder (GNOS) development and

manufacture project led by NSSC, CAS. The research at WEGC was supported by the Austrian

Aeronautics and Space Agency of the Austrian Research Promotion Agency (FFG-ALR) under



projects OPSCLIMVALUE (grant No. 848013) and EOPCLIMTRACK (grant No. 859773).
The ECMWF (Reading, UK) is thanked for access to their archived analysis and forecast data
(available at http://www.ecmwf.int/en/forecasts/datasets) and NOAA-NCEI (Boulder, CO, USA)
for access to their radiosonde data archive (available at http://www.ncdc.noaa.gov/data-access
/weather-balloon-data).

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




**Table 1.** Main parameters of the FY-3C GNOS instrument. (Source: Bai et al., 2014; Liao et al., 2016)

| Parameters | Content |
|---|---|
| **GNSS signals** | GPS L1, L2; BDS B1, B2 |
| **Channel numbers** | Positioning: 8; Occultation: 6 (GPS) 4 (BDS) |
| **Sampling rate** | Positioning & ionosphere occultation: 1 Hz<br>Atmosphere occultation: Close loop 50 Hz,<br>Open loop 100 Hz |
| **Output observations** | Type: L1C/A, L2C, L2P/ B1I, B2I<br>Contents: Pseudo-range/carrier phase/SNR |
| **Clock stability** | $1 \times 10^{-12}$ (1 sec Allan) |
| **The peak gain of occultation antenna** | 10 dBi (azimuth ±35 °, horizontal ±7.5 °) |
| **Pseudo-range precision** | ≤30 cm |
| **Carrier-phase precision** | ≤2 mm |
| **Real-time positioning precision** | ≤10 m |
| **Total weight** | ≤14 kg |
| **Total power** | ≤40 W |
| **Equivalent noise temperature** | 250 K |
| **Signal bandwith** | GPS P code 20.46 MHz<br>BDS 4.092 MHz |



**Figure captions**

**Figure 1.** Timeline of FY-3 series satellites. The FY-3 series satellites are the Chinese second generation polar-orbiting meteorological satellites including AM, PM, EM and Rainfall types of satellites, which have been/will be launched in three batches. (Source: Sun et al., 2017)

**Figure 2.** Design (functional block diagram) of GNOS instrument. (Source: Du et al., 2016)

**Figure 3.** Overview diagram of the elements of the FY-3C GNOS mission and its data processing and data product levels, from raw data (Level-0) down to retrieved atmospheric and ionospheric
profiles (Level-2).

**Figure 4.** Distribution of the FY3 C-GNOS radio occultation events on 2 October 2013. There were 94 rising BDS RO events (up-looking red triangle), 90 setting BDS RO events (down-looking red triangle), 287 rising GPS RO events (up-looking blue triangle), and 256 setting GPS RO events
(down-looking blue triangle). (Source: Wang et al., 2015)
**Figure 5.** Refractivity deviation from the ECMWF reanalysis for FY-3C GNOS GPS (from 1 November to 31 December 2013). The left panel shows the mean bias (black) and the standard deviation (red), and the right panel shows the samples used vs. altitude. (Source: Liao et al., 2016)

**Figure 6.** Refractivity deviation from the ECMWF reanalysis for FY-3C GNOS BDS. The description is the same as the Figure 5. (Source: Liao et al., 2016)

**Figure 7.** Comparison of NmF2 measurements from GNOS GPS (left panel) / BDS (right panel) occultation and ionosonde data. The linear regression of the absolute NmF2 values was computed
using a standard difference technique, in which the black line is y=x, the red line is the fitted regression, Corr. Coef. is the correlation coefficient, Bias is the statistical bias, and Std is the standard deviation. (Source: Dai et al., 2017b)

**Figure 8.** Comparison of the anomaly correlation coefficients of the without-GNOS-RO-data
reference case (black line) and the other cases, assimilated GPS GNOS RO data only (red line), and assimilated GPS and BDS GNOS RO data (blue line). The left and right panels show the results of northern and southern hemispheres, respectively.

**Figure 9.** Evaluation score card of GPS and BDS FY-3C GNOS RO data assimilation effects on the
GRAPES forecast model.

**Figure 10.** Comparison of average NmF2 values from 17 ionosonde stations and GNOS ionosphere data in the region of magnetic inclinations between 40° and 80° in the northern hemisphere. (Source: Dai et al., 2017b)



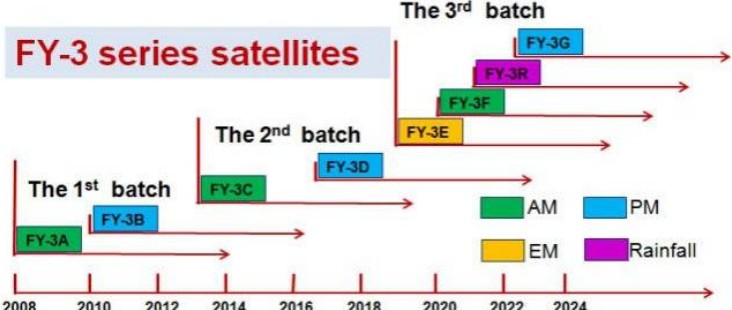

**Figure 1.** Timeline of FY-3 series satellites. The FY-3 series satellites are the Chinese second generation polar-orbiting meteorological satellites including AM, PM, EM and Rainfall types of satellites, which have been/will be launched in three batches. (Source: Sun et al., 2017)

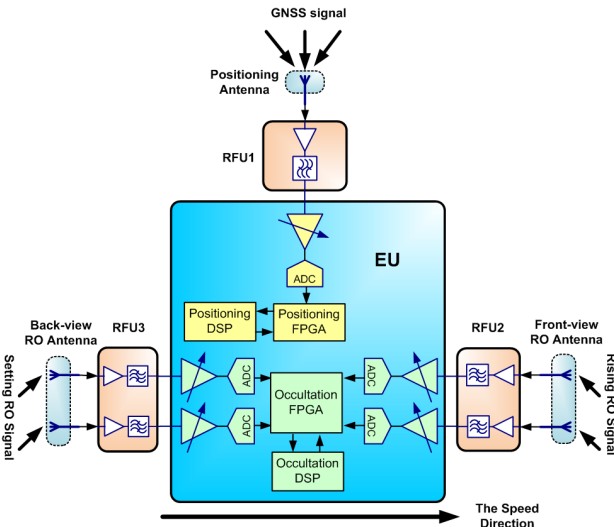

10        **Figure 2.** Design (functional block diagram) of GNOS instrument. (Source: Du et al., 2016)





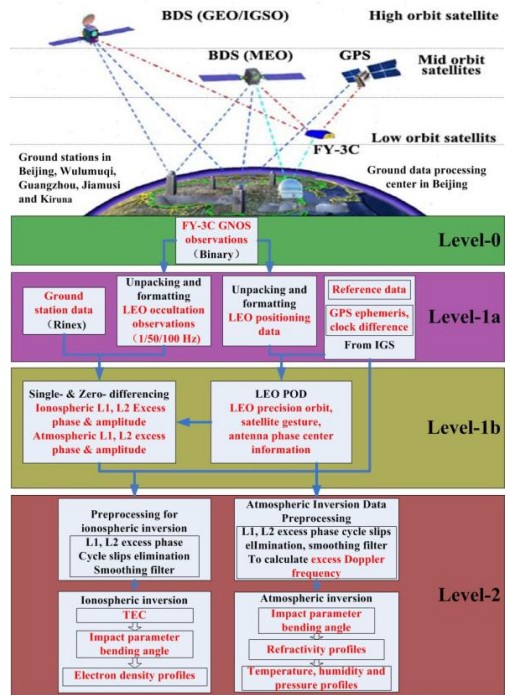

**Figure 3.** Overview diagram of the elements of the FY-3C GNOS mission and its data processing and data product levels, from raw data (Level-0) down to retrieved atmospheric and ionospheric profiles (Level-2).

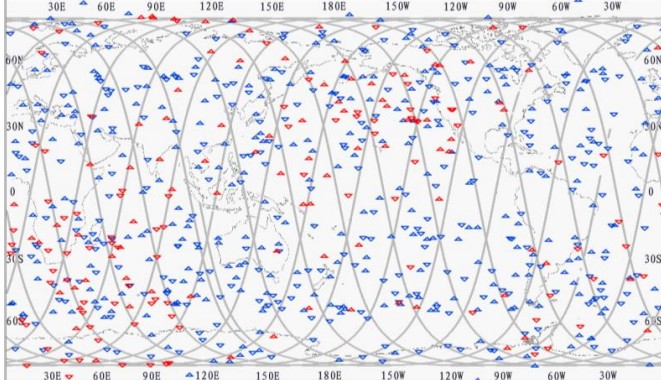

**Figure 4.** Distribution of the FY3 C-GNOS radio occultation events on 2 October 2013. There were 94 rising BDS RO events (up-looking red triangle), 90 setting BDS RO events (down-looking red triangle), 287 rising GPS RO events (up-looking blue triangle), and 256 setting GPS RO events (down-looking blue triangle). (Source: Wang et al., 2015)




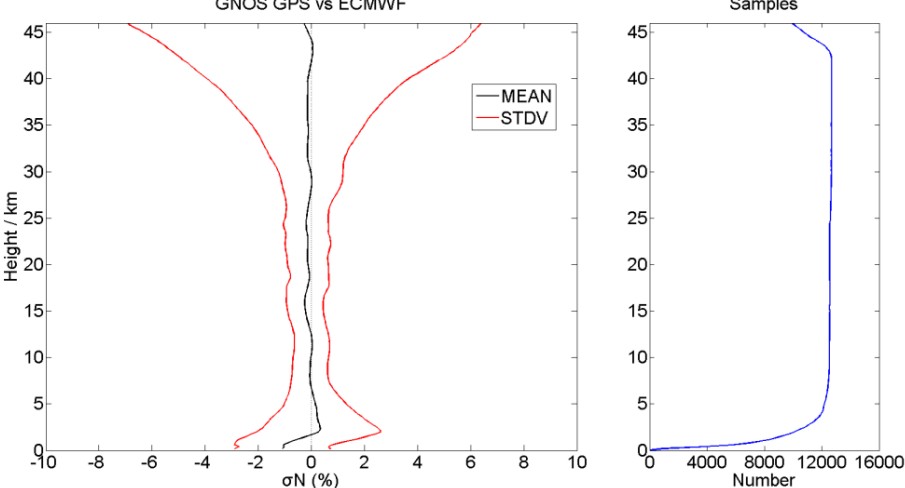

**Figure 5.** Refractivity deviation from the ECMWF reanalysis for FY-3C GNOS GPS (from 1 November to 31 December 2013). The left panel shows the mean bias (black) and the standard deviation (red), and the right panel shows the samples used vs. altitude. (Source: Liao et al., 2016)

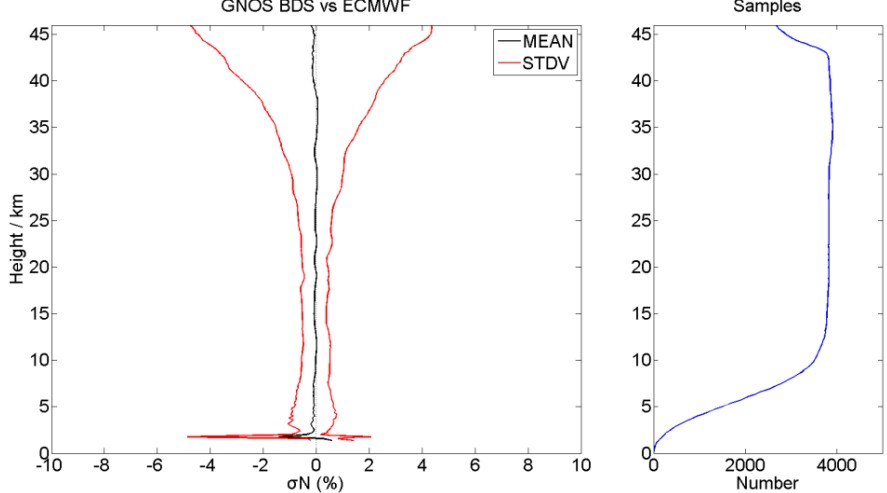

**Figure 6.** Refractivity deviation from the ECMWF reanalysis for FY-3C GNOS BDS. The description is the same as the Figure 5. (Source: Liao et al., 2016)





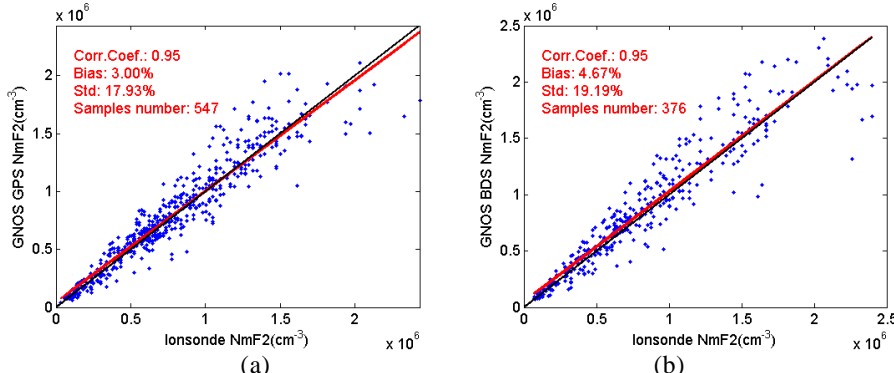

<p style="text-align:center">(a)            (b)</p>

**Figure 7.** Comparison of NmF2 measurements from GNOS GPS (left panel) / BDS (right panel) occultation and ionosonde data. The linear regression of the absolute NmF2 values was computed using a standard difference technique, in which the black line is y=x, the red line is the fitted regression, Corr. Coef. is the correlation coefficient, Bias is the statistical bias, and Std is the standard deviation. (Source: Dai et al., 2017b)

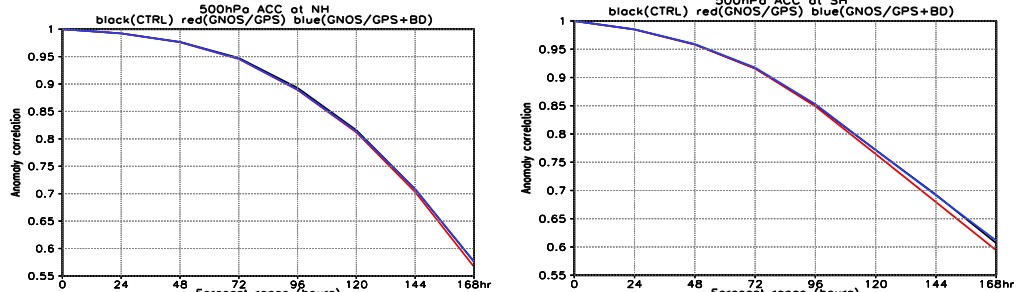

**Figure 8.** Comparison of the anomaly correlation coefficients of the without-GNOS-RO-data reference case (black line) and the other cases, assimilated GPS GNOS RO data only (red line), and assimilated GPS and BDS GNOS RO data (blue line). The left and right panels show the results of northern and southern hemispheres, respectively.





**Figure 9.** Evaluation score card of GPS and BDS FY-3C GNOS RO data assimilation effects on the GRAPES forecast model.



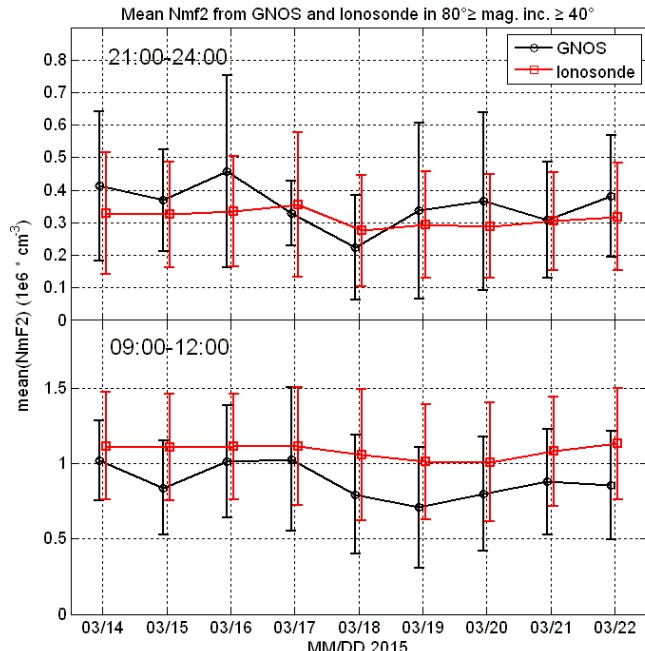

**Figure 10.** Comparison of average NmF2 values from 17 ionosonde stations and GNOS ionosphere data in the region of magnetic inclinations between $40°$ and $80°$ in the northern hemisphere. (Source: Dai et al., 2017b)

