# Peer review of "The FengYun-3C radio occultation sounder GNOS: a review of the mission and its early results and science applications"

_Atmospheric Measurement Techniques, 2017_

## Referee Comment (RC1) · Anonymous Referee #2 · 6 Feb 2018

**Review of paper "The FengYun-3C radio occultation sounder GNOS: a review 1 of the mission and its early results and science applications" by Yueqiang Sun et al.**

The paper reviews the Global Navigation Satellite System (GNSS) occultation sounder (GNOS), data processing, data quality evaluation, and research applications. The data validation demonstrates that the FY-3C GNOS mission can provide atmospheric and ionospheric RO profiles of a quality that is reasonably good for numerical weather prediction (NWP), global climate monitoring (GCM) and space weather research (SWR). The paper can be published after minor revisions, mostly consisting in providing statements that are more accurate, as well as missing definitions.

Page 8: What is the motivation for the 100 Hz sampling rate in open loop? Is it optimal?

Page 12: *The radio occultation processing package (ROPP) software (V6.0) developed at ROM SAF (radio occultation meteorology satellite application facility) is used for this purpose. More specifically, from the excess phase the Doppler frequency can be obtained, then the bending angles are determined from the Doppler frequency shift and the corresponding satellite positions and velocities (e.g., Kursinski et al., 1997).*

Does not ROPP utilize the technique based on Fourier Integral Operators?

M. E. Gorbunov and K. B. Lauritsen, Analysis of wave fields by Fourier Integral Operators and its application for radio occultations, Radio Science, 2004, 39(4), RS4010, doi:10.1029/2003RS002971.

Page 13: *However, there are some higher-order ionospheric effects that still remain in the bending angle profiles (Kursinski et al., 1997; Liu et al., 2013, 2015, 2016, 2017a). To reduce the ionospheric residual errors and other small-scale noise, the statistical optimization technique is used together with the MSISE-90 climatology model. An optimal linear combination is expressed as a matrix equation to compute the neutral atmospheric bending angle and the ionospheric bending angle.*

The term "optimal linear combination" was used be M.Gorbunov. See:

M. E. Gorbunov, Ionospheric correction and statistical optimization of radio occultation data, Radio Science, 2002, 37(5), 17-1–17-9, doi: 10.1029/2000RS002370.

Provide more detail on your optimal linear combination. In what terms is the "matrix equation" formulated?

Page 14: *When GNSS signals transmitted through the ionosphere from GNSS satellites to the FY-3C 14 satellite are bent and delayed …*

It is better to say that bent are the signal propagation paths rather than the signals themselves.

Page 14, Eq. (5). Add some comments on accuracy of this inversion. How large are effects due to horizontal gradients, and the contribution of the ionospheric layers above the LEO.

Page 16–17: *The statistical BDS and GPS GNOS RO data analyses, by using 17 pairs of 22 BDS/GPS GNOS RO events in a week, showed that the BDS/GPS difference standard deviation of refractivity, temperature, humidity, pressure and ionospheric electron density are lower than 2 %, 2 K, 1.5 g/kg, 2 %, and 15.6 %, respectively. Therefore, the BDS observations/products are in general consistent with those from GPS (Wang et al., 2015).*

Are there any systematic differences?

Page 18: How can you explain the difference between the lower-tropospheric bias structures in Figures 5 and 6?

Page 17 and 19: "mean bias" and "average bias" should probably refer to the same quantity. Is it defined as systematic difference averaged over a height interval? If so, such a quantity is not very informative. More interesting is the maximum bias.

Page 20: Define NmF2 (maximum electron density in F2 layer).

Page 22: Define hmF2 (the height of the F2 maximum).

Page 21: The black lines in Figure 8 can hardly be seen under the blue lines. Consider using a different representation, e.g. differences of anomaly correlations.

Page 21–22: *Figure 9 shows an evaluation score card of the effects of the GPS and BDS FY-3C GNOS RO data … better … worse"*

Is it GPS that is better/worse than BDS? The difference between "Far better/worse" and "better/worse" can hardly be seen. Provide the definition of "far" and "not significant".

---

## Referee Comment (RC2) · Anonymous Referee #3 · 8 Feb 2018

The paper provides an overview of the Radio Occultation mission on board the FengYun 3C satellite. It shows a summary of the system architecture and instrument characteristics, processing characteristics, results/validation of neutral atmospheric products (basically refractivity), ionospheric profiles (but only some results on the estimated NmF2 is provided here), on the applications of derived products (thus applications related to assimilation into NWP models, ionospheric products).

The paper is quite well structured, but it is a summary of something presented in other more detailed papers. Nice to have a summary, but the summary should include all the aspects. In this paper, and in particular in the sections related the discussion of prod-

ucts and their validation, only very few examples are provided. For the atmospheric profiles, the discussion is done only at the refractivity level (nothing is said about bending angles and their validation results, which are also very important in the RO community). For the ionosphere monitoring, it is only provided a scatter plot with estimation of correlation between NmF2 derived by processing GNOS data and ionosonde data.

All the results presented are taken by other papers (a reference is always provided), which contains a lot of other interesting information worth to be presented in a summarizing paper like this one.

This is the most critical point I'd like to address to the authors. In this form I'd reject the paper, encouraging the authors to submit a more complete one.

Then there are other major points that I'd like to put in evidence.

Sect 2.4: here you provided some hints on the Geometric Optics (GO) approach to estimate bending angles. But you are using the ROPP software, where also a more efficient wave optics (WO) approach is implemented. Not clear why you provided details on the GO one only. Are you using also the WO retrieval in the lower troposphere or not?

Sect 2.4: regarding TEC estimation. Eq 4 provides you the uncalibrated TEC. For two reasons: first, using only L1-L2, the effect of initial ambiguities is not removed. The TEC is thus completely biased. You should level it to the P2-P1 pseudorange based TEC; second, the leveled TEC should then be corrected by the receiver and transmitter differential code biases. I don't see any description of this standard way to process ionospheric observations.

Sect 2.4: always on the retrieval of ionospheric data. One problem in using the Abel inversion to obtain Ne(h), Eq 5, is the initialization at the LEO height. You should have an estimate of TEC at the LEO height. Could you discuss this?

Sect 3.1: Please use tables to summarize results/validation of POD results. The entire

section presents results in a way that is really difficult to be followed. Moreover, being this a summary, I'd like to see some results also regarding LEO velocities and clock bias estimations.

Sect 3.2: the same here. Use tables to present the results. Define clearly what is the background (true) for evaluating your relative errors. And show/discuss results also at the bending angle level

Sect 3.3: same here. Use tables and try to be more complete in the description of results obtained related the ionospheric monitoring.

Sect. 4.2: I don't understand the difference between this section and Sect 3.3. Here you should have been discussed Application of ionospheric products. Probably something more is done or will be done regarding space weather activities.

I found some typos and bad written sentences. Here a (not exhaustive) list (# shows the row number)

page 2, #10-16: reformulate, it is bad written

page 8, #6: is the open loop baseband signal sampled at 100 kHz or 100 Hz as stated in Table 1?

page 10, #8: are the ground stations used also for computing GPD/BDS orbits and clocks?

page 12, #4-8: bad written. Could you please clarify is you are using single or zero differencing? If BDS and GNOS clock stability is enough, why you have to use single differencing?

page 11, #15: What are GNSS position package data and satellite precision ephemeris data? Bad written. Why you insisted with using "precise ephemeris"? Maybe you wanted to say "precise positions and velocities"?

page 20, # 17: could you please provide further details or define better this ACC?

---

## Author Comment (AC1) · 10 Jun 2018

The comment was uploaded in the form of a supplement:
https://www.atmos-meas-tech-discuss.net/amt-2017-385/amt-2017-385-AC1-
supplement.pdf

---

## Author Comment (AC2) · 10 Jun 2018

Manuscript doi: 10.5194/amt-2017-385, 2017

**Manuscript Title: The FengYun-3C radio occultation sounder GNOS: a review of the mission and its early results and science applications**

Authors: Yueqiang Sun, Weihua Bai, Congliang Liu, Yan Liu, Qifei Du, Xianyi Wang, Guanglin Yang, Mi Liao, Zhongdong Yang, Xiaoxin Zhang, Xiangguang Meng, Danyang Zhao, Junming Xia, Yuerong Cai, and Gottfried Kirchengast

We thank the referee very much for the constructive comments and recommendations. We thoroughly considered all comments and carefully revised the manuscript accounting for most of them. In addition, we carefully complemented these revisions with a couple of further improvements throughout the manuscript text in the spirit of the comments.

Please find below our point-by-point response (in form of italicized, blue text) to the referees' comments (in form of upright, black text), inserted below each comment. Line numbers used in our responses refer to the original AMT Discussions paper and text updates in the revised manuscript are quoted below with yellow highlighting.

**Response to Anonymous Referee #3's Comments**

**Anonymous Referee #3**

**Received and published: 8 February 2018**

The paper provides an overview of the Radio Occultation mission on board the FengYun 3C satellite. It shows a summary of the system architecture and instrument characteristics, processing characteristics, results/validation of neutral atmospheric products (basically refractivity), ionospheric profiles (but only some results on the estimated NmF2 is provided here), on the applications of derived products (thus applications related to assimilation into NWP models, ionospheric products).

The paper is quite well structured, but it is a summary of something presented in other more detailed papers. Nice to have a summary, but the summary should include all the aspects. In this paper, and in particular in the sections related the discussion of products and their validation, only very few examples are provided. For the atmospheric profiles, the discussion is done only at the refractivity level (nothing is said about bending angles and their validation results, which are also very important in the RO community).For the ionosphere monitoring, it is only provided a scatter plot with estimation of correlation between NmF2 derived by processing GNOS data and ionosonde data. All the results presented are taken by other papers (a reference is always provided), which contains a lot of other interesting information worth to be presented in a summarizing paper like this one. This is the most critical point I'd like to address to the authors. In this form I'd reject the paper, encouraging the authors to submit a more complete one.

Thank you for your constructive comments and suggestions, we have revised this paper according to your suggestions and the state-of-the-art status of FY-3C GNOS mission and related publications. It should be a relative complete review paper now; for some interesting points, which still cannot meet your requirements, they are our ongoing research work and they will be explained and published in separate papers.

As you know, FY-3C GNOS is a relatively new GNSS RO mission launched in September

2013. On the one hand, its data validation and scientific application studies have been done and relevant papers have been published. On the other hand, so far there is no review paper like this one to give an overall introduction of FY-3C GNOS mission. Therefore, be think it is the right time to write this review paper to introduce FY-3C GNOS mission, which will help reader to know FY-3C GNOS mission more comprehensively and in a "one-stop-shop paper" in all its aspects.

For the major comments of (1) the atmospheric profiles; (2) the ionosphere monitoring; and (3) All the results presented are taken by other papers (a reference is always provided), our responses are as following:

(1) You are right, bending angles and their validation results are very important, so they are discussed in the revised manuscript now.

(2) Indeed, the GNSS RO technique can provide both TEC and ionospheric electron density product. However, the RO ionosphere data are validated mainly by the comparison analysis of NmF2 with that from ionosonde data, because there is no proper reference electron density profile. So far, the NmF2 also is used to validate COSMIC ionosphere products as a main index. Please refer to the following references:

Hu L. H., Ning B. Q., Liu L. B., Zhao B. Q., Chen Y. D., Li G. Z. 2014, Comparison between ionospheric peak parameters retrieved from COSMIC measurement and ionosonde observation over Sanya, Adv. Space Res., 54, 929-938.

Krankowski A., Zakharenkoiva I., Krypiak-Gregorczyk, A., Shagimura- tov, I.I., Wielgosz, P. 2011, Ionospheric electron density observed by FORMOSAT-3/COSMIC over the European region and validated by ionosonde data, Journal of Geodesy, 85, 949-964.

Lei J., Syndergaard S., Burns A. G., et al. 2007, Comparison of COSMIC Ionospheric Measurements with Ground-Based Observations and Model Predictions: Preliminary Results, Journal of Geophysical Research, 112, 1-6.

(3) Actually, we wrote this paper mainly according to the state-of-the-art status of FY-3C GNOS mission and related publications. To give an overview of FY-3C GNOS mission, we did not only summarize its related papers, but also describe its system architecture, instrument characteristics, data processing characteristics, and our ongoing studies like FY-3C GNOS data NWP applications, which we think are worthy to be published.

All in all, we consider this paper should be published in the interest of the community, to give readers an overview of FY-3C GNOS mission in difference aspects, and also some new knowledge that could not be got from other papers.

Then there are other major points that I'd like to put in evidence.

Sect 2.4: here you provided some hints on the Geometric Optics (GO) approach to estimate bending angles. But you are using the ROPP software, where also a more efficient wave optics (WO) approach is implemented. Not clear why you provided details on the GO one only. Are you using also the WO retrieval in the lower troposphere or not?

Yes, both the GO and WO have been used in our data processing, via the ROPP software. Specifically, the GO has been used above 25 km, while the WO has been used below 25 km. Now, the WO approach also has been described in the revised paper.

Sect 2.4: regarding TEC estimation. Eq 4 provides you the uncalibrated TEC. For two reasons:

first, using only L1-L2, the effect of initial ambiguities is not removed.

The TEC is thus completely biased. You should level it to the P2-P1 pseudorange based TEC; second, the leveled TEC should then be corrected by the receiver and transmitter differential code biases. I don't see any description of this standard way to process ionospheric observations.

You are right. We just retrieve the ionosphere electron density as final GNOS ionospheric product. The relative TEC in our data processing is only a intermediate variable but not product, so the ambiguities and DCB are not removed; we did not calculated the absolute TEC.

Sect 2.4: always on the retrieval of ionospheric data. One problem in using the Abel inversion to obtain Ne(h), Eq 5, is the initialization at the LEO height. You should have an estimate of TEC at the LEO height. Could you discuss this?

The orbit of FY-3C satellite is 833 km height, and we get the ionosphere information above the LEO orbit by using the exponential extrapolation method. Our simulation indicated that the ionosphere above the LEO satellite effect the accuracy of the retrieved results slightly, but less than 0.1%. Therefore, the ionosphere above the LEO satellite can be neglected in the FY-3C GNOS case with its relatively high orbit altitude.

Sect 3.1: Please use tables to summarize results/validation of POD results. The entire section presents results in a way that is really difficult to be followed. Moreover, being this a summary, I'd like to see some results also regarding LEO velocities and clock bias estimations.

Ok, done. Please refer to the revised paper.

Sect 3.2: the same here. Use tables to present the results. Define clearly what is the background (true) for evaluating your relative errors. And show/discuss results also at the bending angle level

The background ("true") data is ECMWF reanalysis data, which has been used as reference data for RO data validation also in other of studies. It is clarified in the revised manuscript now.

A preliminary comparison study of GNOS GPS raw bending angles with COSMIC and MetOp RO data has been conducted; and the results also presented in Liao, et al., 2016 AMT paper. We have input the bending angle valuation results and discussions in the revised manuscript, for details, please refer to Liao, et al., 2016 AMT paper.

Sect 3.3: same here. Use tables and try to be more complete in the description of results obtained related the ionospheric monitoring.

Ok, done. Please refer to the revised paper.

Sect. 4.2: I don't understand the difference between this section and Sect 3.3. Here you should have been discussed Application of ionospheric products. Probably something more is done or will be done regarding space weather activities.

Right, Sect. 3.3 is on validation of the ionosphere data, while Sect. 4.2 describes GNOS

ionosphere data application for a geomagnetic storm monitoring and analysis.

I found some typos and bad written sentences. Here a (not exhaustive) list (# shows the row number)

page 2, #10-16: reformulate, it is bad written *Ok, done*.

page 8, #6: is the open loop baseband signal sampled at 100 kHz or 100 Hz as stated in Table 1?

It is 100 Hz, thank you. Revised.

page 10, #8: are the ground stations used also for computing GPD/BDS orbits and clocks? The ground stations are mainly used to receive the observed data from FY-3C satellite, and then transmit it to the data processing center in Beijing. And their functions are clarified in the revised manuscript now.

page 12, #4-8: bad written. Could you please clarify is you are using single or zero differencing? If BDS and GNOS clock stability is enough, why you have to use single differencing?

Currently, all the GPS/GNOS RO events and part of the BDS/GNOS RO events use singledifferencing algorithm. For the other part of the BDS/GNOS RO events, which do not have a proper BDS reference satellites to implement the single-differencing, we use the zero-differencing algorithm. As we know, the single-differencing algorithm is commonly used in different RO data centers, and for different RO missions, so we mainly use it in operational data processing system.

We conducted a comparison study/analysis of the single- and zero- differencing to validate whether the zero-differencing can be used for the BDS/GNOS RO events that do not have a proper BDS reference satellite.

Thank you for your valuable suggestion; reprocessing our data by using zero-differencing is part of our next plans and on-going work.

page 11, #15: What are GNSS position package data and satellite precision ephemeris data? Bad written. Why you insisted with using "precise ephemeris"? Maybe you wanted to say "precise positions and velocities"?

The GNSS position package we use contains the raw pseudo range and carrier phase observations from positioning antenna, the satellite precision ephemeris data is in GPS SP3 format, which can be obtained from IGS stations. These raw data were used as inputs to calculate the precise positions and velocities.

page 20, # 17: could you please provide further details or define better this ACC?

An approach to measure the quality of a forecast system is to calculate the correlation between forecasts and observations. However, correlating forecasts directly with observations or analyses may give misleadingly high values because of the seasonal variations. It is therefore established practice to subtract the climate average from both the forecast and the verification and to verify the forecast and observed anomalies according to the anomaly correlation coefficient (ACC), which in its most simple form can be written:

$$ACC = \frac{\overline{(f-c)(a-c)}}{\sqrt{(f-c)^2(a-c)^2}}$$

The ACC can be regarded as a skill score relative to the climate. It is positively orientated, with increasing numerical values indicating increasing "success".

We thank the reviewer again for the valuable comments that helped to improve the paper.

---

## Author Comment (AC3) · 10 Jun 2018

Manuscript doi: 10.5194/amt-2017-385, 2017
Manuscript Title: **The FengYun-3C radio occultation sounder GNOS: a review of the mission and its early results and science applications**
Authors: Yueqiang Sun, Weihua Bai, Congliang Liu, Yan Liu, Qifei Du, Xianyi Wang, Guanglin Yang, Mi Liao, Zhongdong Yang, Xiaoxin Zhang, Xiangguang Meng, Danyang Zhao, Junming Xia, Yuerong Cai, and Gottfried Kirchengast

*We thank the referee very much for the constructive comments and recommendations and for the overall positive rating that this is considered a useful paper clearly worthy of publication. We thoroughly considered all comments and carefully revised the manuscript accounting for most of them. In addition, we carefully complemented these revisions with a couple of further improvements throughout the manuscript text in the spirit of the comments.*
*Please find below our point-by-point response (in form of italicized, blue text) to the* referees' comments (in form of upright, black text)*, inserted below each comment. Line numbers used in our responses refer to the original AMT Discussions paper and text updates in the revised manuscript are quoted below with* yellow highlighting*.*

**Response to Anonymous Referee #1's Comments**

The paper reviews the Global Navigation Satellite System (GNSS) occultation sounder (GNOS), data processing, data quality evaluation, and research applications. The data validation demonstrates that the FY-3C GNOS mission can provide atmospheric and ionospheric RO profiles of a quality that is reasonably good for numerical weather prediction (NWP), global climate monitoring (GCM) and space weather research (SWR). The paper can be published after minor revisions, mostly consisting in providing statements that are more accurate, as well as missing definitions.
*Thank you. Agreed, and the statements and definitions were improved.*

Page 8: What is the motivation for the 100 Hz sampling rate in open loop? Is it optimal?
*Currently, for the sampling rate in open loop, COSMIC is 50 Hz and MetOp is 1000 Hz (raw sampling rate). According to Sokolovskiy et al, studies, for open loop, 50 Hz sampling rate is sufficient to monitor the troposphere, while MetOp uses 1000 Hz sampling rate to do a detailed spectrum analysis of the lower troposphere. Indeed, the sampling rate is the higher the more detailed information of atmosphere can be obtained, at least from 50 Hz to 1000 Hz. FY-3C GNOS adopts 100 Hz sampling rate because the limitation of the FY-3C satellite down link load capability. 100 Hz is the highest sampling rate we can use for FY-3C GNOS, so we can draw the conclusion that 100 Hz is optimal for FY-3C GNOS.*
*For the detailed information of open loop sampling rate analysis, please refer to the following references:*
*Sokolovskiy, S.: Tracking tropospheric radio occultation signals from low Earth orbit, Radio Sci., 36, 483–498, 2001.*
*Sokolovskiy, S. V., Rocken, C., Lenschow, D. H., Kuo, Y.-H., Anthes,R. A., Schreiner, W. S., and Hunt, D. C.: Observing the moist troposphere with radio occultation signals from COSMIC, Geophys. Res. Lett., 34, L18802, doi:10.1029/2007GL030458, 2007.*

*Sokolovskiy, S., Rocken, C., Schreiner, W., Hunt, D. C., and Johnson, J.: Postprocessing of L1 GPS radio occultation signals recorded in open-loop mode, Radio Sci., 44, RS2002, doi:10.1029/2008RS003907, 2009*

Page 12: The radio occultation processing package (ROPP) software (V6.0) developed at ROM SAF (radio occultation meteorology satellite application facility) is used for this purpose. More specifically, from the excess phase the Doppler frequency can be obtained, then the bending angles are determined from the Doppler frequency shift and the corresponding satellite positions and velocities (e.g., Kursinski et al., 1997).

Does not ROPP utilize the technique based on Fourier Integral Operators?

M. E. Gorbunov and K. B. Lauritsen, Analysis of wave fields by Fourier Integral Operators and its application for radio occultations, Radio Science, 2004, 39(4), RS4010, doi:10. 1029/2003RS002971.

*Right, in FY-3C GNOS data processing, we use the technique based on Fourier Integral Operators, through the ROPP software. Now, the wave-optics (WO) method has been clarified and the M. E. Gorbunov 2004 paper has been cited in the revised manuscript.*

Page 13: However, there are some higher-order ionospheric effects that still remain in the bending angle profiles (Kursinski et al., 1997; Liu et al., 2013, 2015, 2016, 2017a). To reduce the ionospheric residual errors and other small-scale noise, the statistical optimization technique is used together with the MSISE-90 climatology model. An optimal linear combination is expressed as a matrix equation to compute the neutral atmospheric bending angle and the ionospheric bending angle.

The term "optimal linear combination" was used be M.Gorbunov. See: M. E. Gorbunov, Ionospheric correction and statistical optimization of radio occultation data, Radio Science, 2002, 37(5), 17-1–17-9, doi: 10.1029/2000RS002370.

Provide more detail on your optimal linear combination. In what terms is the "matrix equation" formulated?

*Right, the optimal linear combination approach that is nested in ROPP software has been used for FY-3C GNOS data processing, and we did not change it. The details of this approach have been described in the abovementioned M. Gorbunov. 2002 paper and ROPP USER GUIDE: PRE-PROCESSOR section 2.4. Since, in this paper, we focus on the FY-3C GNOS missoin and its data validation and application, so we cited this publication as a reference in the revised manuscript now.*

Page 14: When GNSS signals transmitted through the ionosphere from GNSS satellites to the FY-3C 14 satellite are bent and delayed ⋯

It is better to say that bent are the signal propagation paths rather than the signals themselves.

*Ok, we agree; done.*

Page 14, Eq. (5). Add some comments on accuracy of this inversion. How large are effects due to horizontal gradients, and the contribution of the ionospheric layers above the LEO.

*The statistics of GNOS NmF2 is in line with CHAMP mission, whose NmF2 average bias is -1.7 %, and standard deviation is 17.8 % (Jakowski et al., 2002).*

Jakowski, N., Wehrenpfennig, A., Heise, S., Reigber, C., Lühr, H., Grunwaldt, L., and Meehan, T. K.: GPS radio occultation measurements of the ionosphere from CHAMP: Early results, Geophysical Research Letters, 29, 95-91-95-94, 10.1029/2001gl014364, 2002.
*We have added this reference in the revised manuscript now.*

*For the effects due to horizontal gradients: a simulation study shown that the horizontal gradients could bring about -6.3 to -10.1% relative error (RMS).*
Wu, X., Hu, X., Gong, X., Zhang, X., and Wang, X.: Analysis of inversion errors of ionospheric radio occultation, GPS Solutions, 13, 231-239, 10.1007/s10291-008-0116-x, 2009.

*For the effect of ionospheric layers above the LEO: the orbit of FY-3C satellite is 833 km height, and our simulation evaluation indicated that the ionosphere above the LEO satellite effect the accuracy of the retrieved results slightly, but less than 0.1%. Therefore, the ionosphere above the LEO satellite can be neglected in FY-3C GNOS situation.*

Page 16–17: The statistical BDS and GPS GNOS RO data analyses, by using 17 pairs of 22 BDS/GPS GNOS RO events in a week, showed that the BDS/GPS difference standard deviation of refractivity, temperature, humidity, pressure and ionospheric electron density are lower than 2 %, 2 K, 1.5 g/kg, 2 %, and 15.6 %, respectively. Therefore, the BDS observations/products are in general consistent with those from GPS (Wang et al., 2015).
Are there any systematic differences?
*In this statistical analysis, we did not find obvious systematic differences between the BDS and GPS occultation data.*

Page 18: How can you explain the difference between the lower-tropospheric bias structures in Figures 5 and 6?
*Figure 5 and 6 show the statistics of FY-3C GNOS/GPS and FY-3C GNOS/BDS RO datasets, respectively. The difference between the low-troposphere bias structures due to FY-3C GNOS/GPS RO uses the open loop while FY-3C GNOS/BDS RO use the close loop techniques at the lower troposphere. Therefore, below 3 km height, the amount of FY-3C GNOS/BDS RO data decreases sharply.*

Page 17 and 19: "mean bias" and "average bias" should probably refer to the same quantity. Is it defined as systematic difference averaged over a height interval? If so, such a quantity is not very informative. More interesting is the maximum bias.
*Thanks. Yes both the "mean bias" and "average bias" refer to the systematic difference averaged over a height interval.*
*We also discussed it with the authors of Liao et al., AMT, 2016 paper. In our opinion, the mean bias and standard deviation are statistically meaningful, which can show the dataset's overall quality. However, the maximum bias may happen in some specific RO events, which cannot demonstrate the overall quality of the GNOS RO observations.*

Page 20: Define NmF2 (maximum electron density in F2 layer).
*Ok, done.*

Page 22: Define hmF2 (the height of the F2 maximum).
*Ok, done.*

Page 21: The black lines in Figure 8 can hardly be seen under the blue lines. Consider using a different representation, e.g. differences of anomaly correlations.
*Yes, the black line is mostly covered (over-shadowed) by the blue line, but at each subplot's right corner the black line appears. Thanks for your suggestion to use differences instead, but we think the original figure is a better format, so we preferred to keep the original figure.*

Page 21–22: Figure 9 shows an evaluation score card of the effects of the GPS and BDS FY-3C GNOS RO data … better … worse"
Is it GPS that is better/worse than BDS? The difference between "Far better/worse" and "better/worse" can hardly be seen. Provide the definition of "far" and "not significant".
*Actually, Figure 9 (currently revised to 10) does not show the comparison of the GPS and BDS GNOS RO data quality, but shows the comparison of the NWP accuracy with and without GNOS RO data. Therefore, in Figure 9 better/worse means the NWP accuracy by using GNOS RO data than that without using the GNOS RO data.*

*We thank the reviewer again for the valuable comments that helped to improve the paper.*